# Non-canonical regulation of the reactivation of an oncogenic herpesvirus by the OTUD4-USP7 deubiquitinases

Shaowei Wang[1,2,3☉], Xuezhang Tian[1,2☉], Yaru Zhou[1,2], Jun Xie[1,2], Ming Gao[1,2], Yunhong Zhong[1,2], Chuchu Zhang[1,2], Keying Yu[2,4], Lei Bai[5], Qingsong Qin[6], Bo Zhong[2,4], Dandan Lin[7], Pinghui Feng[8], Ke Lan[2,5], Junjie Zhang[1,2,3]*

**1** State Key Laboratory of Oral & Maxillofacial Reconstruction and Regeneration, Key Laboratory of Oral Biomedicine Ministry of Education, Hubei Key Laboratory of Stomatology, School & Hospital of Stomatology, State Key Laboratory of Virology, Medical Research Institute, Wuhan University, Wuhan, China, **2** Frontier Science Center for Immunology and Metabolism, Wuhan University, Wuhan, China, **3** Hubei Key Laboratory of Tumor Biological Behaviors, Hubei Province Cancer Clinical Study Center, Zhongnan Hospital of Wuhan University, Wuhan, China, **4** Department of Gastrointestinal Surgery, Medical Research Institute, Zhongnan Hospital of Wuhan University, Wuhan University, Wuhan, China, **5** State Key Laboratory of Virology, School of Life Sciences, Wuhan University, Wuhan, China, **6** Laboratory of Human Virology and Oncology, Shantou University Medical College, Shantou, China, **7** Cancer Center, Renmin Hospital of Wuhan University, Wuhan University, Wuhan, China, **8** Section of Infection and Immunity, Herman Ostrow School of Dentistry, Norris Comprehensive Cancer Center, University of Southern California, Los Angeles, California, United States of America

☉ These authors contributed equally to this work.
* junjiezhang@whu.edu.cn

**Data Availability Statement:** All relevant data are within the manuscript and its Supplemental Information files.

## Abstract

Deubiquitinases (DUBs) remove ubiquitin from substrates and play crucial roles in diverse biological processes. However, our understanding of deubiquitination in viral replication remains limited. Employing an oncogenic human herpesvirus Kaposi's sarcoma-associated herpesvirus (KSHV) to probe the role of protein deubiquitination, we found that Ovarian tumor family deubiquitinase 4 (OTUD4) promotes KSHV reactivation. OTUD4 interacts with the replication and transcription activator (K-RTA), a key transcription factor that controls KSHV reactivation, and enhances K-RTA stability by promoting its deubiquitination. Notably, the DUB activity of OTUD4 is not required for K-RTA stabilization; instead, OTUD4 functions as an adaptor protein to recruit another DUB, USP7, to deubiquitinate K-RTA and facilitate KSHV lytic reactivation. Our study has revealed a novel mechanism whereby KSHV hijacks OTUD4-USP7 deubiquitinases to promote lytic reactivation, which could be potentially harnessed for the development of new antiviral therapies.

## Author summary

Kaposi's sarcoma-associated herpesvirus (KSHV) is an oncogenic herpesvirus associated with several human malignancies. The life cycle of KSHV comprises latency and the lytic phase. The transition from latency to the lytic phase, known as reactivation, is crucial for viral replication and pathogenesis. Previous studies have established that KSHV

**Funding:** J.Z. is supported by the National Key Research and Development Program of China (2021YFC2701800, 2021YFC2701804), a startup fund from Wuhan University, grants from National Natural Science Foundation of China (82372241, 31970156, and 82172261), and the Fundamental Research Funds for the Central Universities (2042022dx0003). D.L. is supported by the National Key Research and Development Program of China (2022YFC3401500). Q.Q. is supported by the Open Research Fund Program of the State Key Laboratory of Virology of China (2022KF010). P.F. is supported by grants from NIH (R35DE027556, R01DE026003, R01CA221521, R21AI134105). The funders had no role in study design, data collection and analysis, decision to publish, or preparation of the manuscript.

**Competing interests:** The authors have declared that no competing interests exist.

reactivation is orchestrated by a viral transcription factor, RTA. However, the specific molecular mechanisms governing RTA stability have remained elusive. In this study, we demonstrate that KSHV RTA recruits two host deubiquitinases (DUBs), OTUD4, and USP7, to promote RTA deubiquitination and stabilization, facilitating viral reactivation. Intriguingly, the DUB activity of OTUD4 is dispensable; instead, OTUD4 serves as an adaptor protein that recruits USP7 to deubiquitinate and stabilize RTA. Our study not only unveils the mechanism by which KSHV exploits the OTUD4-USP7 deubiquitinases to enhance lytic reactivation but also provides insights into the development of novel anti-viral therapeutic strategies targeting host deubiquitinase complexes.

## Introduction

The ubiquitination system plays a key role in controlling the stability of target proteins and regulating a variety of cellular events, such as autophagy, DNA damage response, and antiviral innate immunity [1–3]. Approximately 700 E3 ligases are responsible for conjugating ubiquitin to target proteins, while only ~100 deubiquitinases (DUBs) exist to remove ubiquitin from substrates, suggesting that DUBs likely employ complex strategies to target specific substrates and biological events. To date, these cellular DUBs have been classified into seven families: ubiquitin-specific proteases (USPs), ovarian tumor proteases (OTUs), ubiquitin C-terminal hydrolases (UCHs), Machado-Josephin domain proteases (MJDs), zinc-finger-containing Ub peptidase (ZUP1), the JAB1/MPN/MOV34 family (JAMMs), and the motif interacting with Ub-containing novel DUB family (MINDYs) [4, 5]. These DUBs regulate a broad range of physiological and pathological processes including viral infection and pathogenesis.

Herpesviruses are highly prevalent in the human population and herpesvirus infection causes significant morbidity and mortality, especially in immunocompromised individuals. Kaposi's sarcoma-associated herpesvirus (KSHV), also known as human herpesvirus 8, is an oncogenic herpesvirus that is etiologically associated with Kaposi's sarcoma (KS), KSHV-associated inflammatory cytokine syndrome (KICS), and two other lymphoproliferative diseases, primary effusion lymphoma (PEL) and multicentric Castleman's disease (MCD) [6–10]. Like other herpesviruses, the life cycle of KSHV consists of two distinct phases: latent and lytic [11]. Following primary infection, KSHV establishes latency to evade host immune detection and facilitate persistent infection [12]. Intriguingly, a small subset of infected cells spontaneously initiate lytic reactivation, which provides essential paracrine signaling for tumorigenesis [13, 14]. Moreover, the released KSHV progeny virions are able to infect the neighboring cells, thereby replenishing the latently infected cell pool [15]. As KSHV reactivation is crucial for viral replication and oncogenesis, viral and cellular players that modulate the switch between the latent and lytic phase have been actively investigated [12,16]. These studies have established that the replication and transcription activator (K-RTA) encoded by KSHV ORF50 functions as a master transcription factor that controls the latent-lytic switch [12,16]. Further studies have revealed that the stability of K-RTA is regulated by the ubiquitin-proteasome pathway. A KSHV-encoded microprotein named vSP-1 associates with K-RTA and prevents K-RTA degradation through the ubiquitin-proteasome pathway to facilitate lytic reactivation [17]. MDM2 has been identified as an E3 ligase for K-RTA that promotes K-RTA ubiquitination and degradation [18]. In addition, host NCOA2 protein competes with MDM2 for K-RTA binding to stabilize K-RTA, thereby promoting lytic reactivation [19]. However, whether and how host deubiquitinases (DUBs) regulate K-RTA stability and control KSHV lytic reactivation remains elusive.

In this study, OTUD4, a member of the OTU DUB family, has been identified as a new K-RTA-binding partner that promotes K-RTA deubiquitination and stabilization to facilitate KSHV lytic reactivation. Interestingly, the DUB activity of OTUD4 is not required to stabilize K-RTA; instead, OTUD4 functions as an adaptor protein to recruit another DUB, USP7, belonging to the USP DUB family, to enhance K-RTA level and promote KSHV lytic replication. A KSHV mutant with K-RTA defective in OTUD4 recruitment showed impaired lytic replication. Together, our study reveals a novel mechanism whereby KSHV hijacks OTU-D4-USP7 deubiquitinases to promote lytic reactivation, which could be potentially harnessed for the development of new antiviral therapies.

## Results

### KSHV K-RTA interacts with OTUD4

To gain insight into host factors that are involved in KSHV reactivation, we purified KSHV K-RTA from HEK293T cells and analyzed the interacting partners of K-RTA by mass spectrometry analysis. Our proteomics approach identified previous K-RTA-interacting proteins including NCOA2 [19], validating our analysis. We found that a host deubiquitinating enzyme (DUB) OTUD4 was associated with K-RTA in our analysis (S1A Fig). Overexpression of HA-OTUD4 interacted with FLAG-K-RTA in HEK293T cells, and endogenous OTUD4 also formed a complex with K-RTA when we induced KSHV lytic reactivation in both SLK.iBAC and BCBL-1 cells (Figs 1A, 1B and S1B). These results confirmed that K-RTA interacts with OTUD4. Next, confocal microscopy analysis indicated that OTUD4 distributed in both cytoplasm and nucleus in KSHV latently infected cells, while a large portion of OTUD4 translocated into the nucleus during KSHV lytic reactivation (Fig 1C). Notably, K-RTA co-localized with OTUD4 in the nucleus during viral lytic replication (Fig 1C). These data indicate that KSHV K-RTA and OTUD4 form a complex during KSHV lytic replication. Interestingly, A C45A mutant that abrogates OTUD4 DUB activity [20,21] still interacted comparably with K-RTA, indicating that the DUB activity of OTUD4 is dispensable for K-RTA binding (Fig 1D).

Next, we mapped the regions that mediate the interaction between K-RTA and OTUD4. Firstly, we generated a series of OTUD4 truncated mutants and found that OTUD4 (1-425aa), which includes the OTU domain, mediated the interaction with K-RTA (Fig 1E). Subsequently, we generated a series of K-RTA mutants and found that the C-terminal transactivation domain (490-691aa) of K-RTA was responsible for associating with OTUD4 (Fig 1F). We further truncated K-RTA (490-691aa) and found that K-RTA (651-691aa) was required for OTUD4 binding (S1C Fig). Moreover, K-RTA (1-676aa) failed to bind with OTUD4 (S1D Fig), suggesting that K-RTA (677-691aa) mediates the binding with OTUD4. Then we performed alanine scanning in K-RTA (677-691aa) and found that triple mutation of the C-terminal three amino acids of K-RTA (F689A/R690A/D691A) abrogated OTUD4 binding (Fig 1G). We also generated the corresponding three single mutants (F689A, R690A and D691A), and found that each single mutation partly diminished OTUD4 binding (S1E Fig). We conclude that the C-terminal FRD motif of K-RTA mediates OTUD4 binding.

Together, these data indicate that K-RTA interacts with OTUD4 and the C-terminal FRD motif of K-RTA and the N-terminal 1-425aa of OTUD4 mediate the interaction.

### OTUD4 promotes KSHV lytic replication independent of its DUB activity

Next, we further characterized the role of OTUD4 in KSHV lytic reactivation. Knockdown of OTUD4 in SLK.iBAC cells significantly impaired KSHV lytic reactivation, as indicated by decreased transcription of an immediate-early gene ORF50, early genes ORF56 and ORF57,

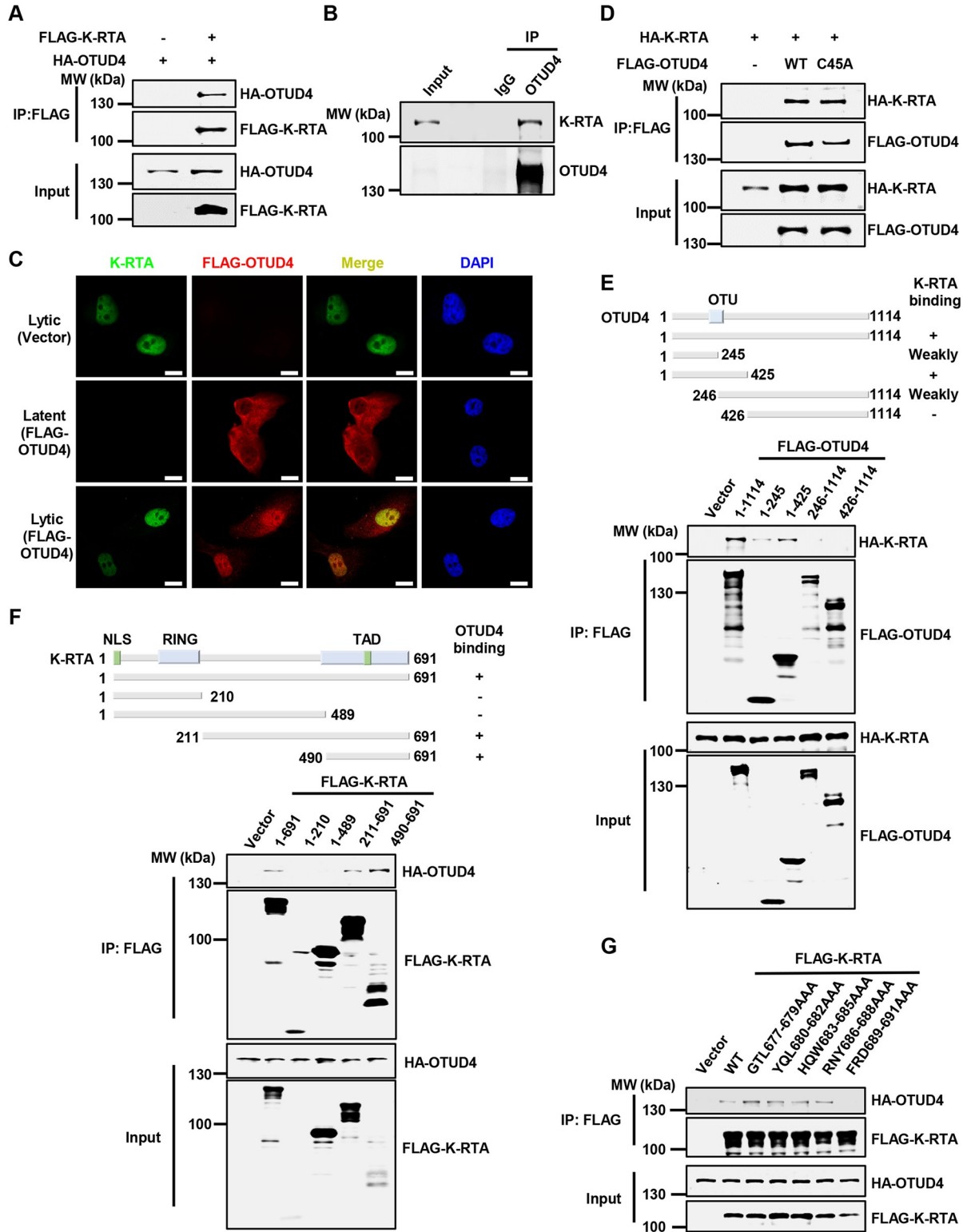

**Fig 1. KSHV K-RTA interacts with OTUD4.** (A) HEK293T cells were transfected with the indicated plasmids, and whole cell lysates (WCLs) were collected for immunoprecipitation with anti-FLAG affinity agarose. The input and precipitated samples were analyzed by immunoblotting. (B) SLK.iBAC-GFP cells were induced with Doxycycline (Dox, 1 μg/ml) for 24 h to trigger lytic reactivation, and co-immunoprecipitation and immunoblotting were performed with the indicated antibodies. (C) SLK.iBAC cells stably expressing vector control or FLAG-OTUD4 were induced with Dox (1 μg/ml) for 24 h, and immunofluorescence was performed using anti-FLAG and K-RTA

antibodies. Scale bars, 20 μm. (D-E) HEK293T cells were transfected with the indicated plasmids, and WCLs were collected for immunoprecipitation with anti-FLAG affinity agarose, followed by immunoblotting. (F) The interaction between K-RTA truncations, including K-RTA(1–210), K-RTA(1–489), K-RTA(211–691), and K-RTA(490–691), and OTUD4 was assessed by co-immunoprecipitation in HEK293T cells. (G) The interaction between K-RTA point mutations, including K-RTA(G677A/T678A/L679A), K-RTA(Y680A/Q681A/L682A), K-RTA(H683A/Q684A/W685A), K-RTA(R686A/N687A/Y688A), and K-RTA(F689A/R690A/D691A), and OTUD4 was assessed by co-immunoprecipitation in HEK293T cells.

and late genes ORF25 and ORF26 (Fig 2A). By contrast, *MAVS*, *ALKBH2*, and *ALKBH3*, which are regulated by OTUD4 at the post-transcriptional level [21, 22], showed no significant transcriptional changes after knock-down of OTUD4 (S2A Fig), indicating that knockdown of OTUD4 does not globally impact cellular transcription activity. Consistently, the protein levels of K-RTA, ORF57, ORF45 and K8α were decreased in the OTUD4-depleted cells (Fig 2B). Moreover, silencing of OTUD4 reduced infectious virion production by around 8-fold (Fig 2C and 2D). Next, we used BCBL-1-Tet-K-RTA, a human primary effusion lymphoma (PEL) cell line that initiates lytic reactivation upon doxycycline induction, and confirmed that knock-down of OTUD4 greatly impaired KSHV lytic replication (Figs 2E, S2B and S2C). Then we asked whether ectopic expression of OTUD4 promotes KSHV reactivation. Indeed, both OTUD4 and the C45A mutant promoted KSHV infectious virion production and enhanced the expression of viral proteins (S2D–S2E Fig). We next performed rescue experiments to rule out the off-target effect of shRNA-mediated knockdown of OTUD4. We reconstituted OTUD4-depleted SLK.iBAC cells with an shRNA-resistant OTUD4 construct or a C45A mutant. While knockdown of OTUD4 markedly repressed KSHV viral titer and viral protein abundance, OTUD4 reconstitution reversed the effect of OTUD4 deficiency on KSHV lytic replication (Fig 2F and 2G). The DUB-deficient mutant C45A also effectively rescued KSHV lytic reactivation (Fig 2F and 2G), confirming that OTUD4 promotes KSHV lytic reactivation independent of its DUB activity. Collectively, these data indicate that OTUD4 promotes KSHV lytic reactivation independent of its DUB activity.

## OTUD4 stabilizes K-RTA by promoting K-RTA K218 deubiquitination

Since K-RTA plays a crucial role in KSHV reactivation and OTUD4 interacts with K-RTA, we hypothesized that OTUD4 may modulate K-RTA protein level to promote lytic reactivation. Indeed, OTUD4 WT or C45A mutant expression enhanced K-RTA expression in a dose-dependent manner (Fig 3A), consist with our previous observations that the DUB activity of OTUD4 is not required for promoting KSHV lytic reactivation. Along the line, OTUD4 WT or C45A mutant expression promoted K-RTA-induced PAN promoter activity in a dose-dependent manner (Fig 3B). Similarly, K-RTA-induced K57/vIL6 promoter activities were also significantly increased when OTUD4 WT or C45A was co-expressed (S3A Fig). To dissect how OTUD4 enhances K-RTA expression, we knocked down OTUD4 and then treated the cells with MG132 to block proteasomal degradation or Bafilomycin A1 to block lysosomal degradation. Knockdown of OTUD4 reduced K-RTA protein level, and MG132 treatment greatly increased K-RTA abundance in both OTUD4-replete and OTUD4-depleted cells. By contrast, Bafilomycin A1 treatment failed to reverse K-RTA downregulation caused by OTUD4 deficiency (S3B Fig). These data suggest that OTUD4 enhances K-RTA protein level by modulating the proteasomal degradation pathway. To further rule out the involvement of autophagy in this process, we took advantage of an HEK293T $ATG7^{-/-}$ cell line that is autophagy-deficient and found that OTUD4 WT and C45A could still enhance K-RTA level in $ATG7^{-/-}$ cells, indicating that autophagy was not involved in OTUD4-mediated K-RTA stabilization (S3C Fig). Next, we asked whether OTUD4 promotes K-RTA deubiquitination and found that both OTUD4 WT and C45A led to K-RTA deubiquitination (Fig 3C). Moreover, both OTUD4 WT

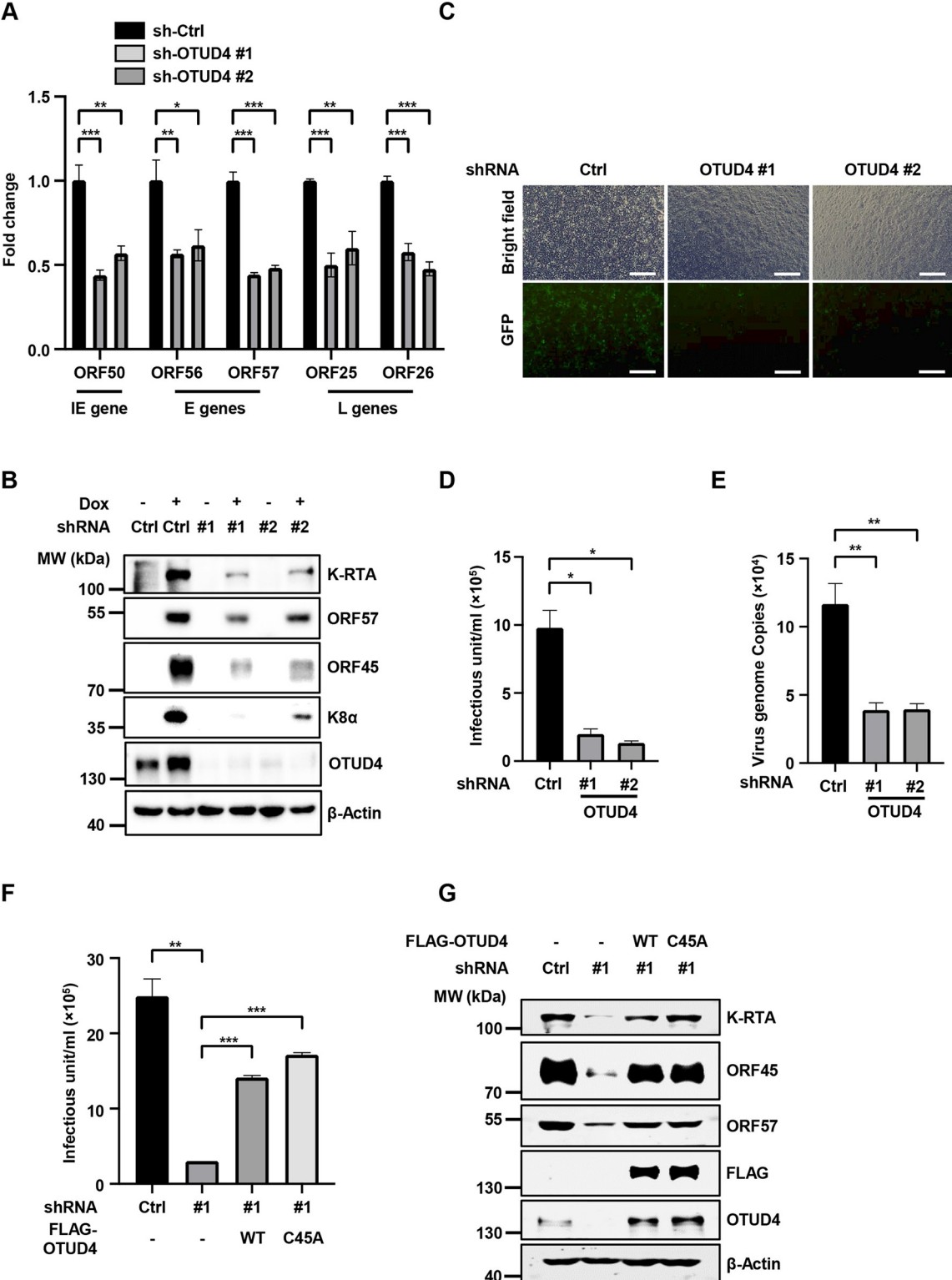

**Fig 2. OTUD4 promotes KSHV lytic reactivation independent of its DUB activity.** (A) SLK.iBAC-GFP cells stably transduced with sh-Ctrl or sh-OTUD4 were induced with Dox (1 μg/ml) for 48 h, followed by quantification of viral gene expression by RT-qPCR. (B) Immunoblotting of SLK.iBAC-GFP cells as described in Fig 2A. (C) SLK.iBAC-GFP cells as described in Fig 2A were induced with Dox (1 μg/ml) and sodium butyrate (0.5 mM) for 48 h. The supernatants containing infectious virion were collected and used to infect HEK293T cells, and GFP expression was imaged at 48 h post-infection. Scale bars, 100 μm. (D) KSHV infectious units were calculated

based on flow cytometry analysis of GFP-positive cell percentage as described in Fig 2C. (E) BCBL1-Tet-K-RTA cells transduced with sh-Ctrl or sh-OTUD4 were induced with Dox (1 µg/ml) and sodium butyrate (0.5 mM) for 48 h. KSHV genome copies in the supernatants were quantified by qPCR. (F) OTUD4 knockdown SLK.iBAC cells were stably reconstituted with control vector, OTUD4 or OTUD4-C45A, and induced with Dox (1 µg/ml) and sodium butyrate (0.5 mM). KSHV infectious units were quantified at 48 h post-induction. (G) SLK.iBAC stable cells as described in Fig 2F was induced with Dox (1 µg/ml) for 48 h, followed by immunoblotting analysis.

and C45A facilitated the removal of K48- but not K63-linked ubiquitination (Figs 3D and S3D). As a control, OTUD4 effectively reduced K63 ubiquitination of Myd88, a known OTUD4 substrate [20], while the C45A mutant exhibited impaired DUB activity (S3E Fig). These data indicate that OTUD4 promotes the removal of K48-linked ubiquitination of K-RTA independent of the DUB activity of OTUD4.

To identify the lysine residues of K-RTA that respond to OTUD4-mediated deubiquitination, we generated two K-RTA truncation mutants covering the C-terminal OTUD4 binding FRD motif (211-691aa and 490-691aa). The protein level of K-RTA (211-691aa) was effectively increased by OTUD4, while the abundance of K-RTA (490-691aa) was not affected by OTUD4 expression, suggesting that K-RTA (211-489aa) consists of critical lysine residues that are critical for OTUD4-mediated K-RTA stabilization (S3F Fig). To further test this, we replaced all 12 lysine residues in 1-210aa with arginine within the full-length of K-RTA and validated that the protein level of the K-RTA-K12R mutant was augmented by OTUD4 (S3F Fig). Then we generated full-length K-RTA mutants in which each of the 7 lysine residues in 211-489aa was individually replaced with arginine. K-RTA reporter activity assays indicated that the activity of the K-RTA K218R mutant did not respond to OTUD4 expression while the activities of the other mutants were enhanced by OTUD4 (Fig 3E). These data suggest that OTUD4 promotes K-RTA K218 deubiquitination to stabilize K-RTA. Consistently, OTUD4 expression promoted the removal of K48-linked ubiquitination of K-RTA and the other 6 K-RTA KR mutants but not the K-RTA K218R mutant (S3G Fig).

Together, these data indicate that OTUD4 stabilizes K-RTA by promoting K-RTA K218 deubiquitination.

## Characterization of a KSHV mutant carrying K-RTA defective in OTUD4 recruitment

Since the C-terminal FRD motif of K-RTA mediates OTUD4 binding, we introduced F689A/R690A/D691A triple mutation (hereafter referred to as K-RTA-3A) into KSHV genome using an infectious bacterial artificial chromosome (BAC) clone (S4A Fig) [23, 24]. In contrast to WT K-RTA, the K-RTA-3A mutant failed to associate with OTUD4 during lytic reactivation (Fig 4A). Consistently, K-RTA-3A mutation significantly impaired viral protein expression and progeny virion production (Figs 4B, 4C and S4B) and knockdown of OTUD4 could not impair the reactivation of KSHV-K-RTA-3A as it did against WT KSHV (S4C and S4D Fig). Moreover, OTUD4/C45A failed to promote the lytic reactivation of KSHV-RTA-3A as it did for WT KSHV (Fig 4D). These data indicate that OTUD4 associates with K-RTA via the C-terminal FRD motif of K-RTA to facilitate KSHV lytic reactivation.

## USP7 promotes KSHV lytic reactivation

Since OTUD4 promotes K-RTA deubiquitination and stabilizes K-RTA independent of the DUB activity of OTUD4, we reason that OTUD4 may recruit another DUB to catalyze the deubiquitination of K-RTA. We analyzed all the binding partners of OTUD4 generated in previous proteomics studies and found three DUBs (USP10, CYLD and USP7) potentially

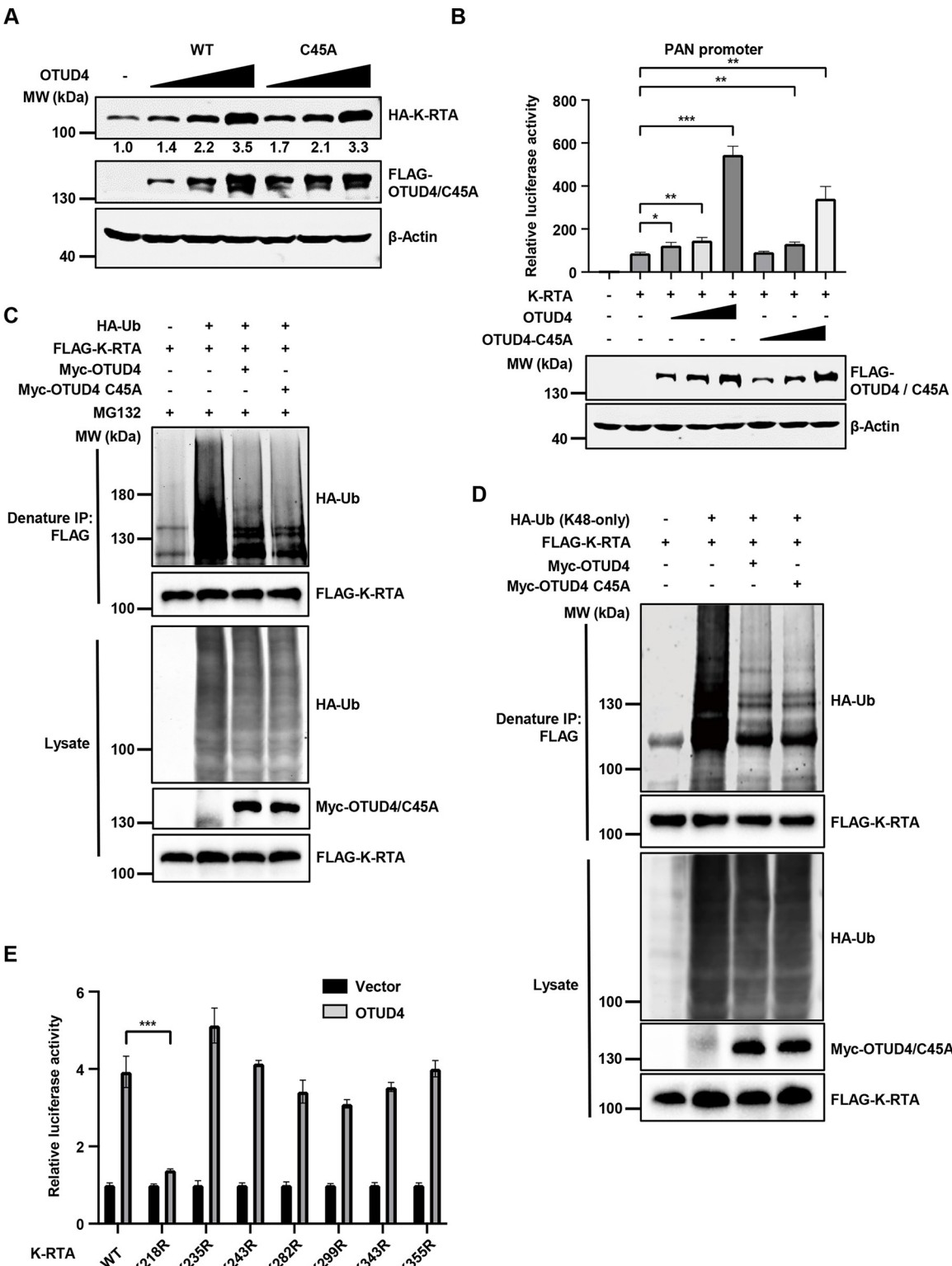

**Fig 3. OTUD4 facilitates K-RTA deubiquitination and stability by promoting K-RTA K218 deubiquitination.** (A) HEK293T cells were co-transfected with HA-K-RTA and different amounts of FLAG-OTUD4/C45A (0, 0.5, 1 or 2 μg). WCLs were collected at 24 h post-transfection and analyzed by immunoblotting. Densiometric analysis of the bands was performed with Image J. (B) HEK293T cells were co-transfected with a luciferase reporter plasmid driven by KSHV PAN promoter (PAN-Luc), HA-K-RTA and different amounts of FLAG-OTUD4 WT/C45A mutant (0, 0.1, 0.2 or 0.5 μg). Luciferase activities were determined at 24 h post-transfection. (C) HEK293T

cells were co-transfected with FLAG-K-RTA, HA-Ub and Myc-OTUD4/C45A, and then treated with MG132 (10 μM). Denatured immunoprecipitation with anti-FLAG affinity agarose was performed, followed by immunoblotting. (D) HA-Ub (K48-only), instead of HA-Ub, was used for the detection of K-RTA ubiquitination as describe in Fig 3C. (E) HEK293T cells were co-transfected with PAN-Luc reporter, vector control or FLAG-OTUD4, and WT K-RTA or the indicated mutants. Luciferase activities were quantified at 24 h post-transfection.

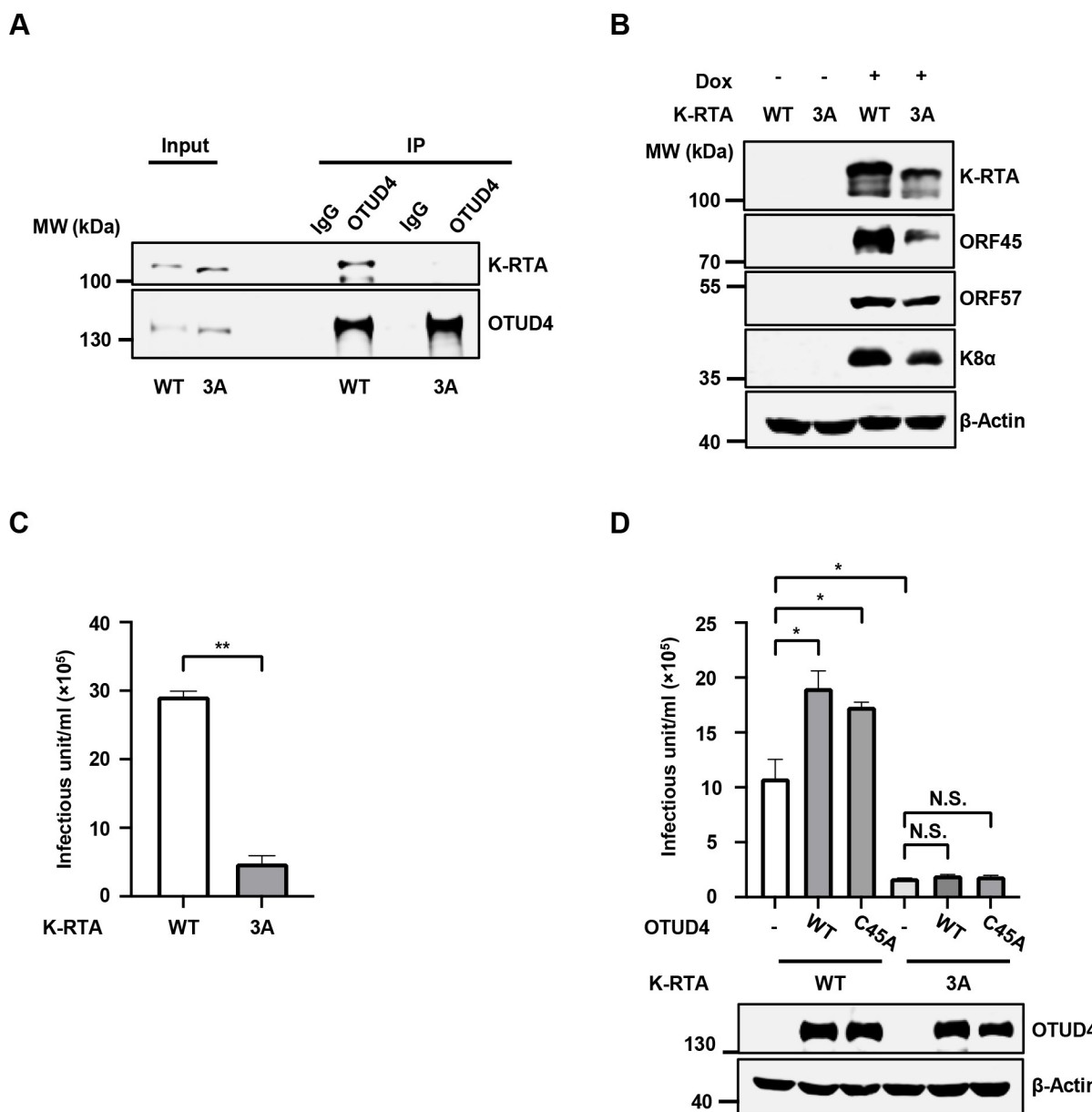

**Fig 4. A KSHV mutant with K-RTA defective in OTUD4 recruitment showed impaired lytic replication.** (A) SLK.iBAC-K-RTA-WT or SLK.iBAC-K-RTA-F689A/R690A/D691A (K-RTA-3A) cells were induced with Dox (1 μg/ml) for 48 h, followed by co-immunoprecipitation and immunoblotting. (B) SLK.iBAC-K-RTA-WT or SLK.iBAC-K-RTA-3A cells were induced with Dox (1 μg/ml) for 48 h, and WCLs were collected and analyzed by immunoblotting. (C) SLK.iBAC-K-RTA-WT or SLK.iBAC-K-RTA-3A cells were induced with Dox (1 μg/ml) and sodium butyrate (0.5 mM) for 48 h, and KSHV infectious units were quantified. (D) SLK.iBAC-K-RTA-WT or SLK.iBAC-K-RTA-3A cells stably transduced with OTUD4 WT/C45A were induced with Dox (1 μg/ml) and sodium butyrate (0.5 mM) for 48 h, and KSHV infectious units were quantified. WCLs were collected and analyzed by immunoblotting.

interacted with OTUD4 [22, 25–27]. Overexpression of USP7, but not USP10 or CYLD, enhanced the protein level of K-RTA (Figs 5A, S5A and S5B). Furthermore, knockdown of USP7, but not USP10 or CYLD, greatly diminished KSHV progeny virion production (Figs 5B and S5C–S5E). These results suggest that USP7, but not USP10 or CYLD, plays a critical role in KSHV lytic reactivation. Then we confirmed that USP7 functions as a DUB and clearly reduced K48 ubiquitination of MDM2 (S5F Fig). Next, we found that overexpression of HA-OTUD4 interacted with FLAG-USP7 in HEK293T cells and endogenous USP7 also formed a complex with OTUD4 when we induced KSHV lytic reactivation in SLK.iBAC and BCBL-1 cells (Figs 5C and S5G). Domain mapping experiments indicated that the N-terminal region of OTUD4 (1-425aa) interacted with USP7 (S5H Fig). GST pull-down assay revealed that the TRAF domain (1-208aa), but not the tandem UBL domain (560-1102aa) of USP7, was able to bind OTUD4 (S5I Fig). Knockdown of USP7 greatly suppressed KSHV lytic replication, as indicated by the reduced transcription of KSHV lytic genes (Fig 5D). By contrast, *MAVS*, *ALKBH2*, and *ALKBH3* showed no significant transcriptional changes after knockdown of USP7 (S5J Fig), indicating that knockdown of USP7 does not globally impact cellular transcription activity. Consistently, viral protein levels were decreased in USP7-depleted SLK. iBAC cells (Fig 5E). Moreover, USP7 knockdown also significantly reduced KSHV genome replication and viral protein expression in BCBL-1-Tet-K-RTA cells (Figs 5F and S5K). These data indicate that USP7 promotes KSHV lytic reactivation. Notably, knockdown of USP7 in OTUD4-depleted cells could not further reduce KSHV progeny virion production (Fig 5G and 5H), suggesting that the OTUD4-USP7 axis plays a critical role in KSHV lytic reactivation.

## OTUD4 bridges the interaction between K-RTA and USP7 to promote KSHV lytic reactivation

Next, we further explored how OTUD4 and USP7 regulate K-RTA level to promote KSHV lytic reactivation. First, we found that K-RTA interacted with USP7 (Figs 6A and S6A). Notably, OTUD4 expression enhanced the interaction between K-RTA and USP7 in a dose-dependent manner (Fig 6B), suggesting that OTUD4 bridges the interaction between K-RTA and USP7. In line with this, knockdown of OTUD4 abolished the interaction between K-RTA and USP7 (Fig 6C), indicating that OTUD4 functions as an adaptor protein between K-RTA and USP7. Moreover, OTUD4/C45A failed to remove K48-linked ubiquitin chains of K-RTA in USP7-depleted cells (S6B and S6C Fig). Functionally, OTUD4 WT or the C45A mutant promoted K-RTA activity in reporter assays, however, the promoting effect was abrogated upon the depletion of USP7 (Fig 6D). Consistently, OTUD4/C45A expression promoted KSHV progeny virion production, while knockdown of USP7 negated the promoting effect of OTUD4/C45A (Fig 6E). Furthermore, USP7 failed to remove K48-linked ubiquitin chains of RTA in OTUD4-depleted cells (Fig 6F). These data indicate that OTUD4 bridges the interaction between K-RTA and USP7 to stabilize K-RTA, thereby promoting KSHV lytic reactivation.

Since we found OTUD4 (1-425aa) is sufficient to associate with both K-RTA and USP7 (Figs 1E and S5H), if OTUD4 functions as an adaptor, we speculate that OTUD4 (1-425aa) would be sufficient to stabilize K-RTA and promote KSHV lytic replication. Consist with this notion, the expression of OTUD4 (1-425aa) strengthened the interaction between K-RTA and USP7 in a dose-dependent manner (Fig 6G). Moreover, OTUD4 (1-425aa) enhanced the protein level of K-RTA and the transcriptional activity of K-RTA, similar to the effects of WT OTUD4 (Figs 6H and S6D). OTUD4 (1-425aa) effectively promoted K-RTA deubiquitination as OTUD4 did (S6E Fig). K-RTA-K218R exhibited decreased ubiquitination compared with K-RTA, and the ubiquitination of K-RTA-K218R remained unchanged when co-expressed

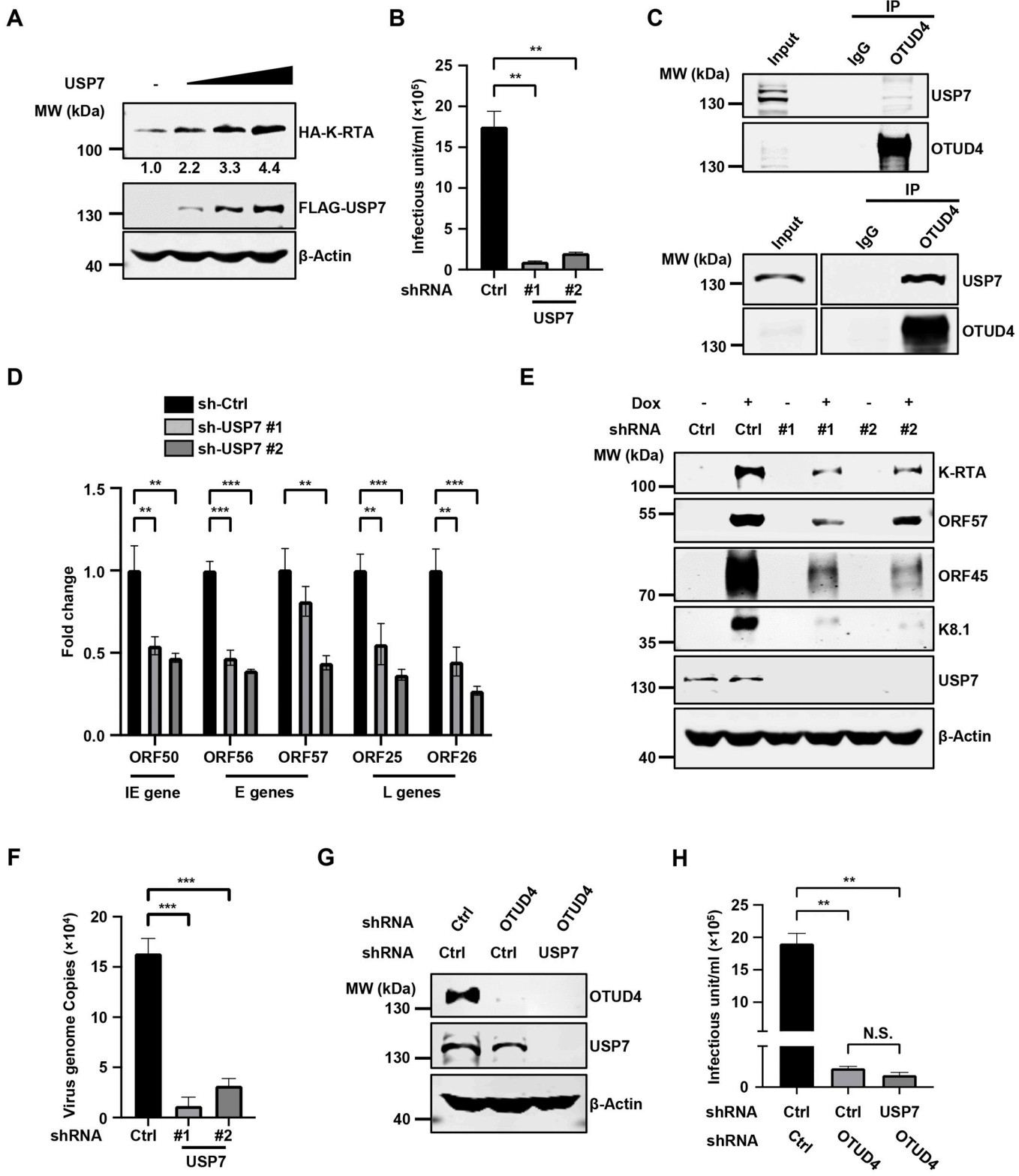

**Fig 5. USP7 interacts with K-RTA and promotes KSHV reactivation.** (A). HEK293T cells were co-transfected with HA-K-RTA and FLAG-USP7 (0, 0.5, 1 or 2 μg), followed by immunoblotting at 24 h post-transfection. (B) SLK.iBAC-GFP cells transduced with sh-Ctrl or sh-USP7 were induced with Dox (1 μg/ml) and sodium butyrate (0.5 mM) for 48 h. The supernatants containing KSHV progeny virion were collected and used to infect HEK293T cells, and KSHV infectious units were calculated based on flow cytometry analysis of GFP-positive cell percentage. (C) SLK.iBAC-GFP was induced with Dox (1 μg/ml) for 24 h to trigger lytic reactivation, and BCBL1-Tet-K-RTA cells were induced with Dox (1 μg/ml) and sodium butyrate (0.5 mM) for 48 h. Co-immunoprecipitation and immunoblotting were performed with the indicated antibodies. (D) SLK.iBAC-GFP cells stably transduced with sh-Ctrl or sh-USP7 were induced with Dox (1 μg/ml) for 48 h, followed by quantification of viral gene expression by RT-qPCR. (E) Immunoblotting of SLK.iBAC-GFP cells as described in Fig 5D. (F) BCBL1-Tet-K-RTA cells transduced with either sh-Ctrl or sh-USP7 were induced with Dox (1 μg/ml) and sodium butyrate (0.5 mM) for 48 h. KSHV genome copies in the supernatants were quantified. (G-H) SLK.iBAC-GFP cells transduced with sh-Ctrl or sh-OTUD4 were further transduced with sh-Ctrl or sh-USP7, and the stable cells were induced with Dox (1 μg/ml) and sodium butyrate (0.5 mM) for 48 h. WCLs were collected and analyzed by immunoblotting (G), and KSHV infectious units were quantified (H).

with OTUD4 and OTUD4 1-425aa (S6E Fig). Reconstitution of OTUD4 (1-425aa) in OTUD4-depleted cells rescued impaired KSHV lytic reactivation, as indicated by increased progeny virion production and enhanced expression of viral proteins (K-RTA and K8α) (Figs 6I and S6F). Finally, we evaluated the dimerization/oligomerization activities of OTUD4 and OTUD4 (1-425aa) and found that both OTUD4 and OTUD4 (1-425aa) could form dimer/oligomer (S6G Fig). These data further support our model whereby OTUD4 forms a dimer/oligomer to facilitate the interaction between K-RTA and USP7, leading to the deubiquitination and stabilization of K-RTA by USP7 to promote KSHV lytic replication.

## Discussion

The ubiquitination-proteasome system plays key roles in controlling the stability of target proteins. Central to this critical process, E3 ubiquitin ligases catalyze the conjugation of ubiquitin to the target proteins, while deubiquitinating enzymes remove ubiquitin chains from the substrates. The ubiquitination-proteasome system is extensively involved in physiological and pathological events including virus infection [28, 29]. As an oncogenic herpesvirus, KSHV has been co-evolving with the host for thousands of years, and accumulating evidence indicates that KSHV hijacks host ubiquitin system to facilitate infection [30]. Of note, KSHV encodes multiple E3 ligases that target a wide range of substrates. For example, KSHV-encoded E3 ligase K-RTA catalyzes IRF7 polyubiquitination and targets IRF7 for proteasomal degradation to dampen antiviral innate immunity [31]. K-RTA also promotes the degradation of the Hey1 repressor protein and SMC5/6 to facilitate lytic replication [32, 33]. KSHV-encoded two membrane-bound ubiquitin E3 ligases, K3 and K5, mediate the ubiquitination and degradation of a long list of substrates [34], with MHC-I being the most studied [23, 35–38]. Moreover, KSHV encodes a DUB ORF64, which is effective to remove both K48- and K63-linked ubiquitin chains [39]. ORF64 catalyzes RIG-I deubiquitination to suppress RIG-I-mediated antiviral innate immune signaling [40].

In addition to KSHV-encoded E3 ligases and DUB, cellular counterparts are also involved in KSHV replication and pathogenesis. MDM2, a ubiquitin E3 ligase, has been found to interact with K-RTA and promote K-RTA ubiquitination and degradation [18]. USP7, a member of the USP DUB family, was originally named herpesvirus-associated ubiquitin-specific protease (HAUSP) since it was first discovered as a strong binding partner of the ICP0 protein of HSV-1 [41, 42]. Early studies reveal that USP7 stabilizes ICP0 to facilitate HSV-1 replication and ICP0 recruits USP7 to antagonize antiviral innate immunity [43–45]. Interestingly, multiple KSHV proteins also interact with USP7, strengthening the critical role of USP7 in herpesvirus infection. KSHV vIRF1 and vIRF4 associate with USP7 to inhibit p53-mediated antiviral responses [46, 47]. Moreover, KSHV ORF45 bridges the binding between USP7 and KSHV ORF33, leading to ORF33 stabilization that is required for infectious virion production [48]. As an OTU DUB family member, ovarian tumor family deubiquitinase 4 (OTUD4) is

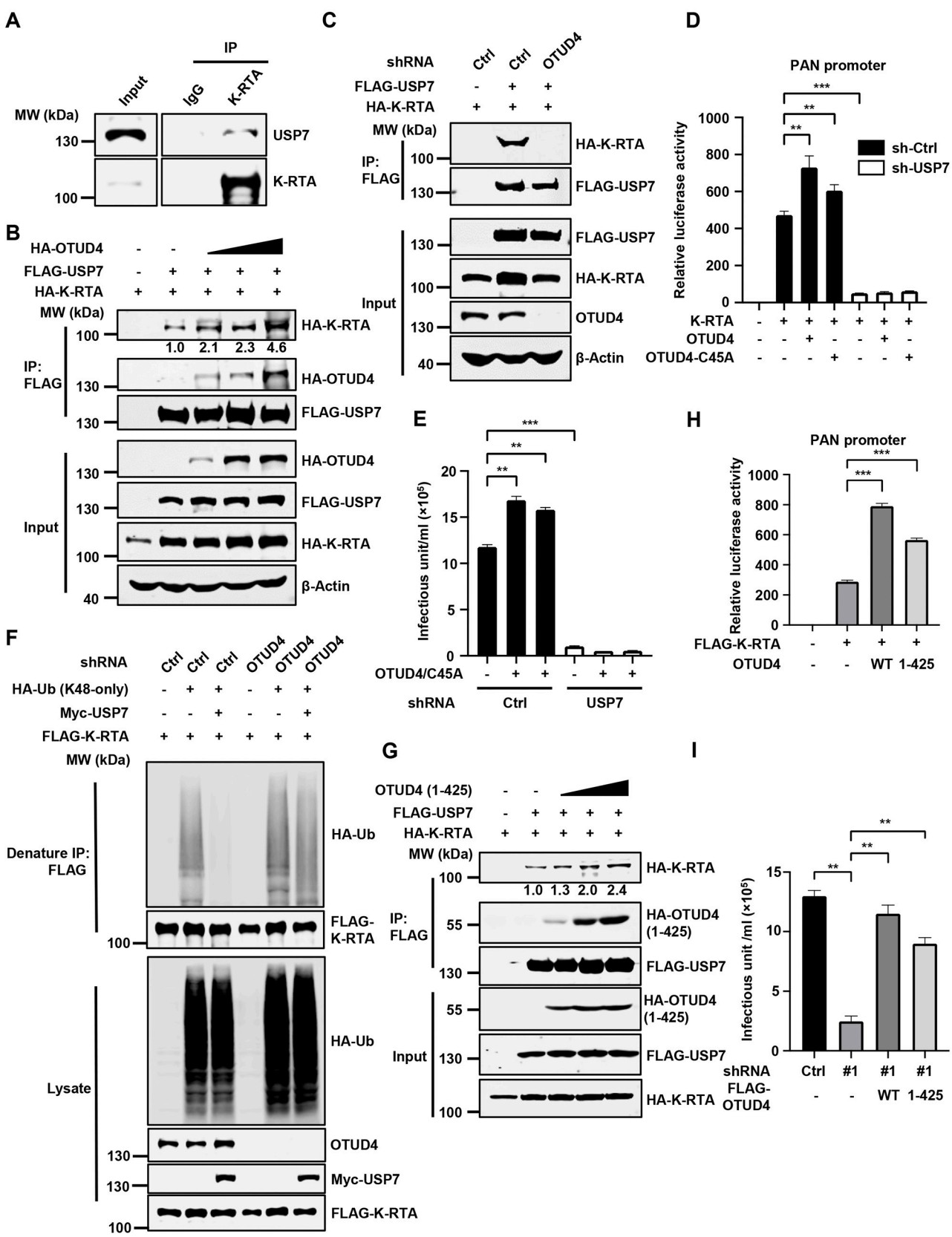

**Fig 6. OTUD4 bridges the association of K-RTA and USP7 to promote KSHV lytic reactivation.** (A) BCBL-1 Tet-K-RTA was induced with Dox (1 μg/ml) for 24 h to trigger lytic reactivation. Co-immunoprecipitation and immunoblotting were performed with the indicated antibodies. (B) HEK293T cells were transfected with the indicated plasmids. WCLs were collected for immunoprecipitation with anti-FLAG affinity agarose, followed by immunoblotting analysis. (C) HEK293T cells transduced with sh-Ctrl or sh-OTUD4 were transfected with the indicated plasmids, followed by immunoprecipitation and immunoblotting analysis at 24 h post-transfection. (D) HEK293T cells transduced with sh-Ctrl or sh-USP7 were co-transfected with PAN-Luc, HA-K-RTA, and FLAG-OTUD4 WT/C45A. Luciferase activities were determined at 24 h post-transfection. (E) SLK. iBAC-GFP cells transduced with sh-Ctrl or sh-USP7 were further stably transduced with vector control, OTUD4 WT or the C45A mutant. Then the stable cells were induced with Dox (1 μg/ml) and sodium butyrate (0.5 mM) to trigger lytic reactivation for 48 h. KSHV infection units in the supernatants were quantified. (F) HEK293T-control or HEK293T-shOTUD4 cells were co-transfected with FLAG-K-RTA, HA-Ub (K48-only) and Myc-USP7, and then treated with MG132 (10 μM). Denatured immunoprecipitation with anti-FLAG affinity agarose was performed, followed by immunoblotting. (G) HEK293T cells were transfected with FLAG-USP7, HA-K-RTA and HA-OTUD4 (1-425aa) (2, 4 or 6 μg). WCLs were collected for immunoprecipitation with anti-FLAG affinity agarose, followed by immunoblotting analysis. (H) HEK293T cells were co-transfected with PAN-Luc, FLAG-K-RTA, and OTUD4 WT or OTUD4 (1-425aa). Luciferase activities were determined at 24 h post-transfection. (I) OTUD4 knockdown SLK. iBAC-GFP cells were stably reconstituted with control vector, OTUD4 or OTUD4 (1-425aa). The stable cells were induced with Dox (1 μg/ml) and sodium butyrate (0.5 mM) to induce lytic reactivation. KSHV infectious units were quantified at 48 h post-induction.

implicated in zebrafish embryonic development and is associated with Ataxia, Dementia, and Hypogonadotropism of human patients [49]. Previous studies reveal that OTUD4 are involved in the regulation of antiviral innate immunity and DNA damage repair caused by DNA alkylation [20–22]. Interestingly, A recent study shows that HSV-1 ICP0 interacts with OTUD4, suggesting that OTUD4 is involved in herpesvirus infection [50]. Here in this study, we present a unique example whereby KSHV hijacks two DUBs, OTUD4 and USP7, to facilitate lytic reactivation. Interestingly, the DUB activity of OTUD4 is not required in this process; instead, OTUD4 functions as an adaptor protein to bridge the interaction between K-RTA and USP7. The non-canonical scaffold function of OTUD4 has been previously observed in DNA repair regulation. OTUD4 has been shown to recruit two deubiquitinases, USP7 and USP9X, which act directly on DNA demethylases ALKBH2 and ALKBH3 to promote DNA alkylation repair [22]. A similar adaptor model has been proposed in innate immune regulation by the deubiquitinase USP18, which recruits another deubiquitinase USP20 to promote innate antiviral response by deubiquitinating STING/MITA [51]. Our domain mapping results indicate that the TRAF domain of USP7 mediated the interaction between USP7 and OTUD4. Previous studies have established that the TRAF domain of USP7 recognizes a conserved P/A/EXXS (X represents any amino acid) motif in its substrate. Currently, we are screening the P/A/EXXS motifs in OTUD4 to determine whether these motifs play critical roles in USP7 binding. Additionally, OTUD4 translocates into the nucleus after KSHV lytic reactivation and co-localizes with K-RTA. However, the mechanism underlying OTUD4 translocation into the nucleus following KSHV lytic reactivation remains unknown. A plausible hypothesis is that K-RTA interacts with OTUD4, facilitating its translocation into the nucleus to promote KSHV lytic reactivation. Further studies are needed to test this hypothesis.

It is worth noting that KSHV vIRF1 and vIRF4 suppress the DUB activity of USP7 to inhibit p53 and promote tumorigenesis [46, 47], whereas USP7 DUB activity is required for K-RTA stability and efficient viral replication during KSHV lytic reactivation. These seeming discrepancies can be explained by two non-mutually exclusive possibilities. Firstly, USP7 may have distinct roles during KSHV latency and lytic replication. vIRF1 and vIRF3 are expressed during the latent phase, and suppression of USP7 DUB activity is critical to inhibit p53 tumor suppressor activity and enable efficient replication of KSHV latently infected cells. By contrast, during KSHV lytic reactivation, the OTUD4-USP7 axis is required to stabilize K-RTA and promote viral replication. Alternatively, during lytic replication, USP7 in the K-RTA-OTUD4-USP7 complex is activated to enable efficient viral replication, while USP7 in the USP7-p53 complex might be specifically targeted by vIRF1 and vIRF3 to negate the antiviral effect of p53 signaling. Since USP7 depletion greatly impairs KSHV lytic reactivation, we reason that USP7 activation in the K-RTA-OTUD4-USP7 axis is favored during lytic reactivation,

although vIRF1 and vIRF3 may still function to suppress USP7 and p53 signaling at the same time. Whether USP7 activity is distinctively modulated in different complexes during KSHV lytic reactivation warrants further study.

Our study sheds light on the development of novel antiviral strategies. Knockout of *Usp7* in mice results in embryonic lethality [52], since USP7 has a wide range of substrates that play pivotal roles in many physiological processes [53]. Therefore, USP7 DUB inhibition as an anti-viral approach may not be feasible due to potential toxicity, despite that a number of potent USP7 small-molecule inhibitors are available [53]. Based on our findings, we propose the development of a peptide or peptide-like inhibitors to disrupt the interaction between OTUD4 and USP7 as a potential antiviral strategy. Such inhibitors could specifically target the OTU-D4-USP7 axis, which is critical for KSHV lytic reactivation while sparing the other vital functions of USP7. As an example, two vIRF4-derived peptides effectively suppress the DUB activity of USP7, leading to p53-dependent cell cycle arrest and tumor regression in mouse models [47]. Whether OTUD4-USP7 and the proposed antiviral strategy affect other herpesviruses including MHV68 and EBV warrants further investigation.

In summary, we report that OTUD4 interacts with K-RTA and enhances K-RTA stability by promoting K-RTA deubiquitination. The DUB activity of OTUD4 is not required for K-RTA stabilization; instead, OTUD4 functions as an adaptor protein to recruit USP7 to deubiquitinate K-RTA and promote KSHV lytic reactivation. Our study has revealed a novel mechanism whereby KSHV hijacks OTUD4-USP7 deubiquitinases to promote lytic reactivation, which could be potentially harnessed to develop new antiviral therapies.

## Materials and methods

### Cell culture

HEK293T cells were cultured in Dulbecco's Modified Eagle Medium (DMEM) (Sigma) supplemented with 10% fetal calf serum (FCS) (Lonsera, Shuangru Biotech, Shanghai, China) and 1% penicillin–streptomycin (HyClone). iSLK cells carrying doxycycline-inducible K-RTA (kindly provided by Drs. Jae Jung, Cleveland Clinic, and Kevin Brulois, Stanford University) were maintained in DMEM supplemented with 10% FCS, 1% penicillin–streptomycin, puromycin (1 μg/ml) and G418 (250 μg/ml) [23]. SLK cell line carrying KSHV clone iBAC (SLK. iBAC-GFP; kindly provided by Dr. Fanxiu Zhu, Florida State University) was maintained in DMEM supplemented with 10% FCS, 1% penicillin–streptomycin and Hygromycin B (500 μg/ml) [24]. BCBL1 cells (kindly provided by Dr. Ke Lan, Wuhan University) were maintained in RPMI-1640 (HyClone) supplemented with 10% FCS and 1% penicillin–streptomycin. BCBL1-Tet-K-RTA cell line was generated by stably transducing BCBL-1 with a doxycycline-inducible K-RTA construct [54].

### Antibodies and other regents

The following antibodies and reagents were used for immunoblotting and immunoprecipitation: Mouse anti-FLAG monoclonal antibody (1:10000; Dia-An Biotechnology, Wuhan, China, catalog no. 2064); Mouse anti-HA monoclonal antibody (1:3000; Dia-An Biotechnology, Wuhan, China, catalog no. 2063); Mouse anti-β-actin monoclonal antibody (1:5000; Dia-An Biotechnology, Wuhan, China, catalog no. 2060); Rabbit anti-K-RTA monoclonal antibody (1:1000, Mabnus biotechnology, Wuhan, China); Mouse anti-KSHV ORF45 monoclonal antibody (1:1000; Santa Cruz, sc-53883); Mouse anti-KSHV ORF57 monoclonal antibody (1:1000; Santa Cruz, sc-135746); Mouse anti-c-Myc monoclonal antibody (1:1000; Santa Cruz, sc-40); Mouse anti-KSHV K8.1A/B monoclonal antibody (1:1000; Santa Cruz, sc-65446); Mouse anti-K8α monoclonal antibody (1:1000; Santa Cruz, sc-69797); Rabbit IgG (Proteintech,

20010049); Rabbit anti-OTUD4 polyclonal antibody (1:1000; Proteintech, 25070-1-AP); Mouse anti-HAUSP/USP7 monoclonal antibody (1:500; Santa Cruz, sc-137008); IRDye 800CW Goat anti-Rabbit and anti-Mouse secondary antibodies (1:10000; LI-COR); Goat anti-Rabbit and anti-Mouse HRP Conjugated secondary antibodies (Bio-Rad); Anti-FLAG magnetic Beads (Bimake, Shanghai, China); Anti-HA magnetic beads (Bimake); Poly FLAG peptide (100 μg/ml; Bimake); Anti-FLAG beads (Dia-An Biotechnology, Wuhan, China); Protein A/G agarose (GE healthcare); Glutathione-Sepharose beads (Smart-lifesciences); Reduced L-glutathione (Solarbio, Beijing, China); DAPI Fluoromount-G mounting medium (Southern-Biotech, 0100–20).

The following antibodies were used for immunofluorescence microscopy (IF): Mouse anti-FLAG monoclonal antibody (1:400; Dia-An Biotechnology, Wuhan, China, #2064); Rabbit anti-K-RTA monoclonal antibody (1:400, Mabnus biotechnology, Wuhan, China); Alexa Fluor 647 conjugated goat anti-mouse secondary antibody (1:1000; Invitrogen #A10680) and Alexa Fluor 594 conjugated goat anti-rabbit secondary antibody (1:1000; Invitrogen #A11012).

Doxycycline (Dox), Formalin and Sodium butyrate were purchased from Sigma-Aldrich; Puromycin, Hygromycin B and G418 were purchased from Invivogen; MG132 was ordered from MedChemExpress; Bafilomycin-A1 (Baf-A1) was purchased from Selleck.

## Plasmids

WT ORF50 and the mutants were sub-cloned into pEF-FLAG-N, pEF-HA-N or pcDNA3.1-FLAG-HA using standard molecular biological technology. pHAGE-OTUD4 WT, C45A mutant and the deletion mutants were kindly provided by Dr. Bo Zhong (Wuhan University) [21]. OTUD4 WT were sub-cloned into pEF-HA-N or pEF-Myc-N, and the point mutations were introduced by site-directed mutagenesis. pRK-FLAG-Myd88 was kindly provided by Drs. Qing Yang and Hong-Bing Shu (Wuhan University). pCMV-FLAG/Myc-USP7 and pEF-FLAG-MDM2 constructs were kindly provided by Dr. Jinfang Zhang (Wuhan University), and USP7 was sub-cloned into pEF-HA-N vector. USP7 (1-208aa) and USP7 (560-1102aa) were subcloned into pGEX-6p-1. HA-Ub, HA-Ub (K48-only) and HA-Ub (K63-only) were kindly provided by Dr. Bo Zhong (Wuhan University). KSHV PAN-, K57- and vIL-6-promoter luciferase reporter plasmids were kindly provided by Dr. Pinghui Feng (University of Southern California) [55, 56]. Short hairpin RNA (shRNA) targeting OTUD4, USP7 and CYLD was constructed into pLKO.1 (Addgene). The following shRNA sequences were used in this study:

shOTUD4 #1: 5′-CAAGTCGAGAATCTAACTATT-3′;
shOTUD4 #2: 5′-TATGCAATGCCTTAGTCATAA-3′;
shUSP7 #1: 5′-CAAGCAGTGCTGAAGATAATA-3′;
shUSP7 #2: 5′-AGCCTGCTACAGACGTTATTT-3′;
shCYLD #1: 5′-GAGGACAGTCTCCGGAATATT-3′;
shCYLD #2: 5′-AGGTTCATCCAGTCATAATAA-3′;
shUSP10 #1: 5′-GCCTTTGAGCCCACATATATT-3′;
shUSP10 #2: 5′-CCTATGTGGAAACTAAGTATT-3′.

## Purification of K-RTA and Mass Spectrometry analysis

HEK293T cells transfected with FLAG-HA-K-RTA were collected and lysed in Nonidet P-40 (NP-40) lysis buffer (150 mM NaCl, 50 mM Tris-HCl, pH 7.4, 1% NP-40, 1 mM EDTA) supplemented with protease inhibitors at 24 h post-transfection. Cell lysates were centrifuged at 13,000 rpm for 10 min at 4°C, and the supernatants were incubated with 50 μl of anti-FLAG magnetic Beads (Bimake) at 4°C for 12 h. Anti-FLAG beads were harvested with a magnetic

separation device, and were then extensively washed with NP-40 lysis buffer. Poly FLAG peptide (100 μg/ml; Bimake) diluted in TBS buffer (150 mM NaCl, 50 mM Tris-HCl, pH 7.4) was used to elute the fusion protein at 4˚C for 6 h. After the first elution, the supernatants were collected and incubated with 50 μl of anti-HA magnetic beads (Bimake) for another 6 h at 4˚C. HA Beads were collected and washed as indicated above. The binding proteins were digested and analyzed by mass spectrometry (BGI Tech, Shenzhen, China). The MS results are summarized in the S1 Table.

## RNA extraction and RT-qPCR

SLK.iBAC-GFP or BCBL1-Tet-K-RTA cells were treated with Dox (1 μg/ml) for 24 h. Total RNA was extracted using TRIzol reagent (Takara) and cDNA was synthesized using HiScript II 1st Strand cDNA Synthesis Kit (Vazyme, Nanjing, China) according to the manufacturer's instructions. Then the cDNA mixture was diluted 40-times and subjected to qPCR analysis with SYBR green qPCR master mix (Bimake, Shanghai, China). The relative quantitation of the target genes was normalized to ATCB. The primer sequences are listed in S2 Table.

## Luciferase reporter assay

Dual-luciferase reporter assay was conducted to evaluate the transcriptional activity of K-RTA [57]. WT or USP7 knockdown HEK293T cells, seeded into 24-well plates, were transfected with 50 ng of TK- renilla luciferase reporter plasmid together with 50 ng of PAN-Luc, K57-Luc or vIL-6-Luc luciferase reporter plasmid, 50ng of HA-K-RTA, and different amounts of FLAG-OTUD4 (100, 200, or 500 ng), OTUD4-C45A (100, 200, or 500 ng) or OTUD4 1-425AA (100, 200, or 500 ng). Luciferase activities were determined 24 h post-transfection using the dual-luciferase reporter assay kit (Vazyme, Nanjing, China).

## Immunoprecipitation

For immunoprecipitation, HEK293T cells were transfected with the indicated constructs and cells were collected and lysed in NP-40 lysis buffer supplemented with a protease inhibitor cocktail at 24 h post-transfection. Cell lysates were centrifuged at 12,000 rpm for 10 min at 4˚C and then the supernatants were incubated with anti-FLAG beads (Dia-An Biotechnology, Wuhan, China) at 4˚C for 4 h. For immunoprecipitation of endogenous proteins, SLK.iBAC-GFP cells were induced with Dox (1 μg/ml) for 24 h and whole cell lysates were prepared as indicated above. Cell lysates were incubated with rabbit IgG control (Proteintech) or the indicated antibodies (2 μg) for 6 h, followed by incubation with 10 μl of protein A/G agarose (GE healthcare) for another 8–10 hours on a roller at 4˚C. FLAG beads or protein A/G agarose were harvested by centrifugation at 4,000 rpm for 2 min at 4˚C, and washed extensively with NP-40 lysis buffer. The binding proteins were released by boiling in SDS sample buffer for 15 min, separated by SDS-PAGE and analyzed by immunoblotting.

## GST pull-down

GST, GST-USP7(1-208aa) or GST-USP7(560-1102aa) were expressed in *E.coli* BL21(DE3) by induction with IPTG (0.5 mM) at 18˚C overnight. The Cells were harvested and sonicated in lysis buffer (20 mM Tris-Cl, pH 8.0, 137 mM NaCl, 2.7 mM KCl, 1% Triton X-100, 0.2 mM PMSF) on ice for 15 min. The supernatant was collected and incubated with Glutathione-Sepharose beads (Smart-lifesciences, Changzhou, China) at 4˚C for 4 h. After extensive washing with wash buffer (20 mM Tris-Cl, pH 8.0, 137 mM NaCl, 2.7 mM KCl, 1% Triton X-100), the bound proteins were eluted with elution buffer (20 mM Tris-Cl, pH 8.0, 137 mM NaCl, 2.7

mM KCl,10 mM reduced L-glutathione) at 4°C for 20 min. The eluted proteins were dialyzed against dialysis buffer (20 mM Tris-Cl, pH 8.0, 150 mM NaCl, 10% glycerol), and incubated with FLAG-OTUD4 expressed in HEK293T cells at 4°C for 4 h in the presence of glutathione beads (20 μl). After washing with wash buffer, glutathione beads were boiled at 95°C for 20 min and the released proteins were analyzed by immunoblotting.

## Ubiquitination assay

HEK293T cells were transfected with FLAG-K-RTA WT or the K-R mutants, HA-Ub (WT, K48-only or K63-only) and Myc-OTUD4 WT, C45A mutant or 1-425aa mutant. After 16 h post-transfection, cells were treated with MG132 (10 μM) for another 10 h. Cells were lysed with 1% Triton X-100 lysis buffer (150 mM NaCl, 50 mM Tris-HCl, pH 7.4, 1% Triton X-100, 1 mM EDTA, with protease inhibitors) supplemented with 1% SDS, and the supernatants were boiled for 5 min to denature proteins. The samples were then diluted 10-fold to reduce the concentration of SDS to 0.1%. Then immunoprecipitation with FLAG beads and immunoblotting were performed as described above.

## Generation of iBAC-K-RTA-3A

K-RTA-3A KSHV-iBAC was generated by two-step Red-mediated recombination as described previously [23,58]. In brief, a linear DNA fragment containing a kanamycin resistance cassette, an I-SceI restriction enzyme site, and the flanking sequences derived from KSHV genome was PCR-amplified with pEP-KanaS as a template. The purified PCR products were electroporated into *E. coli* GS1783 strain containing KSHV iBAC to introduce the first round of recombination. Integration of the Kan/Isce-I cassette was verified by restriction digestion of the extracted BAC and PCR followed by Sanger sequencing. The GS1783 strain containing the modified KSHV iBAC was then incubated with 1% L-arabinose (Sigma) to induce the expression of I-SceI, followed by the second round of recombination to eliminate the Kan/I-SceI cassette. Kanamycin-sensitive and chloramphenicol resistant colonies were picked, and KSHV iBAC was exacted. The successful introduction of the mutation was confirmed by restriction digestion of the extracted iBAC and PCR amplification of the mutated region followed by Sanger sequencing.

The following PCR primers were used in the study:

GTCCGGCACACTGTACCAGCTGCACCAATGGCGTAATTACGCTGCGGCCTGA AGTGTTCGCAAGGGCGT AGGATGACGACGATAAGTAGGG;

GGAAGTTAACGCAGGCACAGACGCCCTTGCGAACACTTCAGgccgcagcGTAATTA CGCCATTGGTGCA AACCAATTAACCAATTCTGATTAG.

## Quantification of KSHV infectious units

SLK.iBAC-GFP cells were treated with Dox (1 μg/ml) and sodium butyrate (0.5 mM) to trigger lytic reactivation. The supernatants were collected at the indicated time points and used to infect HEK293T cells at appropriate dilutions. Then the infected cells were collected, fixed and subjected to flow cytometry analysis. Flow cytometry data were analyzed with FlowJo 10.0, and KSHV infectious units were quantified based on the GFP-positive cell percentage.

## Quantification of viral genome copy number

Quantification of KSHV genome copy number was carried out as previously described [54]. In brief, BCBL1-Tet-K-RTA were treated with Dox (1 μg/ml) and sodium butyrate (0.5 mM) for 48 h to induce lytic reactivation, and the supernatant (500 μl) was collected and treated with

7.5 U of DNase I (Solarbio, Beijing, China) for 1 h at 37°C. Then 30 μl of proteinase K (20 mg/ml, Solarbio, Beijing, China) and 50 μl of 20% SDS was added into the mixture. After incubation at 65°C for 1 h, genomic DNA was extracted by phenol-chloroform extraction and the DNA pellet was resolved in 50 μl of TE buffer. The genomic DNA was diluted 20 times and KSHV genomic DNA was quantified by qPCR. A stand curve was generated using serial dilutions of a pEF-FLAG-K-RTA plasmid. The primers used for the quantification are provided in S2 Table.

### Generation of stable cell lines

Lentiviruses containing the indicated genes or shRNA were generated in HEK293T cells [59]. SLK.iBAC-GFP, BCBL1, iSLK or HEK293T were infected with the indicated lentiviruses, and the transduced cells were selected with puromycin (1 μg/ml) for 2 days after 48 h post-infection. For the reconstitution experiments, SLK.iBAC-GFP OTUD4 knockdown cells were infected with lentiviruses containing vector control, OTUD4 WT, OTUD4-C45A or the OTUD4 1-425aa mutant. The stable cells were maintained in culture medium containing hygromycin B (500 μg/ml) or puromycin (1μg/ml).

For the generation of SLK.iBAC-GFP stable cells that containing indicated mutants of K-RTA, BAC DNA (10μl, ~2 μg) was transfected into SLK cells seeded in 6-well plate ($1.5 \times 10^5$) using 5 μl of Fugene HD (Promega). After 48 h post-transfection, cells were transferred to T25 flask and cultured in the presence of DMEM supplemented with 1% penicillin–streptomycin, 10% FCS and 500 μg/ml Hygromycin B. Then the cells were sub-cultured into appropriate plate until GFP positive colony amplified sufficiently.

### Immunofluorescence

SLK.iBAC-GFP cells were induced with Dox (1 μg/ml) for 24 h to trigger lytic reactivation. The cells were then washed three times with PBS and fixed with 4% (w/v) paraformaldehyde (PFA) (#DF0131, LEAGENE, Beijing, China) for 10 min. Next, the fixed cells were washed with PBS for three times, permeabilized with 1% Triton X-100 for 5 min, and blocked with 10% goat serum (Antgene, Wuhan, China) for 1 hour at room temperature. After washing with wash buffer (1X PBS with 0.1% Tween 20) for three times, the cells were incubated with rabbit anti-K-RTA or mouse anti-FLAG antibodies diluted in PBS containing 1% BSA at 4°C overnight. Cells were washed with wash buffer (PBS containing 0.05% Tween-20) and then incubated with Alexa Fluor 647 conjugated goat anti-mouse secondary antibody (1:1000; Invitrogen) and Alexa Fluor 594 conjugated goat anti-rabbit secondary antibody (1:1000; Invitrogen) at room temperature for 1 hour. After washing with wash buffer and ultrapure water, the slides were mounted with DAPI Fluoromount-G mounting medium (SouthernBiotech). Finally, the images were acquired with a laser scanning confocal microscopy (Leica Stellaris 5) and processed using Image J and Leica image browser [60].

### Statistical analysis

GraphPad Prism (version 8) was used to analysis all statistical data. Data represent the mean of at least three independent experiments, and error bars denote standard deviation (S.D.). Two-tailed student's t test or analysis of variance (ANOVA) was used for statistical analysis. Significant differences are represented by p value (*$p < 0.05$, **$p < 0.01$, ***$p < 0.001$).

### Supporting information

**S1 Fig. KSHV K-RTA interacts with OTUD4.** (A) Affinity purification followed by mass spectrometry analysis to identify K-RTA binding proteins. The number of identified peptides

corresponding to K-RTA, NCOA2, and OTUD4 was summarized. (B) BCBL1-Tet-K-RTA cells were induced with Dox (1 μg/ml) and sodium butyrate (0.5 mM) for 48 h, and co-immunoprecipitation and immunoblotting were performed with the indicated antibodies. (C-E) HEK293T cells were transfected with the indicated plasmids, and WCLs were collected for immunoprecipitation with anti-FLAG affinity agarose, followed by immunoblotting. The interaction between K-RTA truncations, including K-RTAΔ490–535, K-RTAΔ536–589, K-RTAΔ590–650, and K-RTAΔ651–691, and OTUD4 was assessed by co-immunoprecipitation in HEK293T cells (C). The interaction between K-RTA truncations, including K-RTA (1–650), K-RTA (1–663), and K-RTA (1–676), and OTUD4 was assessed by co-immunoprecipitation (D). The interaction between K-RTA point mutations, including K-RTA(F689A), K-RTA (R690A), and K-RTA(D691A), and OTUD4 was assessed by co-immunoprecipitation (E). (TIF)

**S2 Fig. OTUD4 promotes KSHV lytic reactivation independent of its DUB activity.** (A) SLK.iBAC-GFP cells stably transduced with sh-Ctrl or sh-OTUD4 were induced with Dox (1 μg/ml) for 48 h, and the expression of the indicated gene was quantified by RT-qPCR. (B) BCBL1-Tet-K-RTA cells stably transduced with sh-Ctrl or sh-OTUD4 were induced with Dox (1 μg/ml) for 48 h, followed by quantification of viral gene expression by RT-qPCR. (C) Immunoblot of BCBL1-Tet-K-RTA cells as described in S2B Fig. (D) SLK.iBAC stably expressing OTUD4 WT or the C45A mutant was induced with Dox (1 μg/ml) for 48 h, followed by immunoblotting analysis. (E) SLK.iBAC-GFP stable cells as described in S2D Fig were induced with Dox (1 μg/ml) and sodium butyrate (0.5 mM) for 48 h. KSHV infectious units in the supernatants were determined. (TIF)

**S3 Fig. OTUD4 facilitates K-RTA deubiquitination and stability by promoting K-RTA K218 deubiquitination.** (A) PAN-, K57- or vIL-6-reporter was co-expressed with HA-K-RTA and FLAG-OTUD4 WT or C45A mutant in HEK293T cells. Luciferase activities were determined at 24 h post-transfection. (B) iSLK cells transduced with sh-Ctrl (Scramble) or sh-OTUD4 were treated with DMSO, Baf-A1 (1 mM) or MG132 (10 μM), followed by Dox (0.2 μg/ml) induction for 12 h. WCL were collected and analyzed by immunoblotting. (C) HEK293T cells transduced with sgRNA targeting ATG7 were co-transfected with K-RTA and FLAG-OTUD4/C45A (0, 0.5, 1 or 2 μg), followed by immunoblotting at 24 h post-transfection. (D) HEK293T cells were co-transfected with FLAG-K-RTA, HA-Ub (K63-only) and Myc-OTUD4/C45A, and then treated with MG132 (10 μM). Denatured immunoprecipitation with anti-FLAG affinity agarose was performed, followed by immunoblotting. (E) HEK293T cells were co-transfected with FLAG-Myd88, HA-Ub (K63-only) and Myc-OTUD4/C45A. Denatured immunoprecipitation with anti-FLAG affinity agarose was performed, followed by immunoblotting. (F) HEK293T cells were co-transfected with HA-OTUD4 and FLAG-K-RTA or the indicated mutants, followed by immunoblotting at 24 h post-transfection. (G) FLAG-K-RTA WT or the indicated mutants were co-expressed with HA-Ub (K48-only) in HEK293T cells. Then denatured immunoprecipitation was performed as described in S3D Fig. (TIF)

**S4 Fig. A KSHV mutant with K-RTA defective in OTUD4 recruitment showed impaired lytic replication.** (A) PCR products amplified from the K-RTA locus of KSHV-K-RTA-3A were analyzed by Sanger sequencing. (B) SLK.iBAC-K-RTA-WT or SLK.iBAC-K-RTA-3A cells were induced with Dox (1 μg/ml) and sodium butyrate (0.5 mM) for 48 h. The supernatants containing infectious virion were collected and used to infect HEK293T cells, and the infected cells were analyzed by flow cytometry at 24 h post-infection. (C) SLK.

iBAC-K-RTA-WT or SLK.iBAC-K-RTA-3A cells were transduced with sh-Ctrl or sh-OTUD4, and WCLs were analyzed by immunoblotting. (D) SLK.iBAC-K-RTA-WT or SLK. iBAC-K-RTA-3A cells transduced with sh-Ctrl or sh-OTUD4 were induced with Dox (1 μg/ml) and sodium butyrate (0.5 mM) for 48 h, and KSHV infectious units were quantified. (TIF)

**S5 Fig. USP7 interacts with K-RTA and promotes KSHV reactivation.** (A-B) HEK293T cells were co-transfected with HA-K-RTA and FLAG-USP10/CYLD (0, 0.5, 1 or 2 μg), and immunoblotting was performed at 24 h post-transfection. (C-D) SLK.iBAC-GFP cells were transduced with sh-Ctrl, sh-USP10 (C) or sh-CYLD (D), and the stable cells were induced with Dox (1 μg/ml) and sodium butyrate (0.5 mM). The expression of the indicated genes was quantified by RT-qPCR, and KSHV infectious units in the supernatants were quantified at 48 h post-induction. (E) SLK.iBAC-GFP cells were transduced with sh-Ctrl or sh-USP7, and the stable cells were induced with Dox (1 μg/ml) and sodium butyrate (0.5 mM) at 48 h post-transduction. The expression of *USP7* was quantified by RT-qPCR. (F) HEK293T cells were co-transfected with FLAG-MDM2, HA-Ub (K48-only) and Myc-USP7. Denatured immunoprecipitation with anti-FLAG affinity agarose was performed, followed by immunoblotting. (G) HEK293T cells were co-expressed with FLAG-USP7 and HA-OTUD4, followed by co-immunoprecipitation and immunoblotting at 24 h post-transfection. (H) HEK293T cells were co-expressed with HA-USP7 and FLAG-OTUD4 or the indicated mutants, and WCLs were collected for immunoprecipitation with anti-FLAG affinity agarose at 24 h post-transfection. The input and precipitated samples were analyzed by immunoblotting. (I) GST or GST fusion proteins [GST-USP7 (1-208aa) or GST-USP7 (560-1102aa)] were incubated with FLAG-OTUD4 expressed in HEK293T cells. The binding fractions were analyzed by immunoblotting, and purified GST and GST fusion proteins were visualized by Coomassie Brilliant Blue staining. (J) SLK.iBAC-GFP cells stably transduced with sh-Ctrl or sh-USP7 were induced with Dox (1 μg/ml) for 48 h, and the expression of the indicated gene was quantified by RT-qPCR. (K) BCBL1-Tet-K-RTA cells transduced with sh-Ctrl or sh-USP7 were induced with Dox (1 μg/ml) for 48 h. WCLs were collected and analyzed by immunoblotting. (TIF)

**S6 Fig. OTUD4 bridges the association of K-RTA and USP7 to promote KSHV lytic reactivation.** (A) WCLs were collected from HEK293T cells transfected with the indicated plasmids and subsequently subjected to immunoprecipitation with anti-FLAG affinity agarose, followed by immunoblotting. (B) Immunoblotting analysis of HEK293T cells transduced with sh-Ctrl or sh-USP7. (C) HEK293T-shUSP7 cells as described in S6B Fig. were co-transfected with FLAG-K-RTA, HA-Ub (K48-only) and Myc-OTUD4/C45A, and then treated with MG132 (10 μM). Denatured immunoprecipitation with anti-FLAG affinity agarose was performed, followed by immunoblotting. (D) HEK293T cells were co-transfected with FLAG-OTUD4 full length (FL) or 1-425aa with HA-K-RTA, followed by immunoblotting at 24 h post-transfection. (E) HEK293T cells were co-transfected with FLAG-K-RTA or FLAG-K-RTA-K218R, HA-Ub (K48-only), and Myc-OTUD4 full length or 1-425aa, and then treated with MG132 (10 μM). Denatured immunoprecipitation with anti-FLAG affinity agarose was performed, followed by immunoblotting. (F) SLK.iBAC-GFP stable cells as described in Fig 6I were induced with Dox (1 μg/ml) for 48 h, and WCLs were analyzed by immunoblotting. (G) FLAG-OTUD4 and HA-OTUD4 or FLAG-OTUD4 (1-425aa) and HA-OTUD4 (1-425aa) were co-expressed in HEK293T cells, and co-immunoprecipitation and immunoblotting were performed at 24 h post-transfection. (TIF)

**S1 Table. Mass spectrometry data.**
(XLSX)

**S2 Table. Primer information.**
(DOCX)

## Acknowledgments

We thank Drs. Hong-Bing Shu, Ming-Ming Hu, Qing Yang, Jinfang Zhang (Wuhan University), Fanxiu Zhu (Florida State University), Jae Jung (Cleveland Clinic) and Kevin Brulois (Stanford University) for reagents. We thank the core facility of the Medical Research Institute at Wuhan University for excellent technical support.

## Author Contributions

**Conceptualization:** Pinghui Feng, Ke Lan, Junjie Zhang.

**Formal analysis:** Jun Xie, Keying Yu, Lei Bai, Qingsong Qin, Bo Zhong, Dandan Lin, Junjie Zhang.

**Funding acquisition:** Qingsong Qin, Junjie Zhang.

**Investigation:** Shaowei Wang, Xuezhang Tian, Yaru Zhou, Jun Xie, Ming Gao, Yunhong Zhong, Chuchu Zhang.

**Methodology:** Keying Yu, Lei Bai, Qingsong Qin, Bo Zhong, Dandan Lin.

**Supervision:** Junjie Zhang.

**Writing – original draft:** Shaowei Wang, Junjie Zhang.

**Writing – review & editing:** Shaowei Wang, Junjie Zhang.

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
