## [Decision Letter · Decision Letter 0]

26 Oct 2023

Dear Dr. Zhang,

Thank you very much for submitting your manuscript "Non-canonical regulation of the reactivation of an oncogenic herpesvirus by the OTUD4-USP7 deubiquitinases" for consideration at PLOS Pathogens. As with all papers reviewed by the journal, your manuscript was reviewed by members of the editorial board and by several independent reviewers. In light of the reviews (below this email), we would like to invite the resubmission of a significantly-revised version that takes into account the reviewers' comments.

We cannot make any decision about publication until we have seen the revised manuscript and your response to the reviewers' comments. Your revised manuscript is also likely to be sent to reviewers for further evaluation.

Sincerely,

Fanxiu Zhu, Ph.D.

Academic Editor

PLOS Pathogens

Patrick Hearing

Section Editor

PLOS Pathogens

Kasturi Haldar

Editor-in-Chief

PLOS Pathogens

orcid.org/0000-0001-5065-158X

Michael Malim

Editor-in-Chief

PLOS Pathogens

orcid.org/0000-0002-7699-2064

Reviewer's Responses to Questions

**Part I - Summary**

Reviewer #1: In this manuscript, Wang et al found that OTUD4 is a novel binding partner of KSHV RTA and facilitates KSHV lytic replication by stabilizing RTA. Interestingly, the DUB activity of OTUD4 is not required for RTA stability, while OTUD4 acts as an adaptor protein to facilitate RTA-USP7 interaction, which removes the K48-linked ubiquitination of RTA.

Reviewer #2: This data intense study described a function of K-Rta-OTU4 interaction in KSHV replications. The OTU domain containing 4 (OTU4) is a Lys63-specific deubiquitinylase and negative regulators of inflammatory or innate signaling pathways via deubiquitinylation of cellular receptors. The authors comprehensively demonstrated that K-Rta binds to OTU4 during reactivation at C-terminal region, and the interaction is responsible for recruitment of USP7 to the K-Rta protein. The authors further showed that recruitment of USP7 is important for increasing the stability of K-Rta via K-Rta de-ubiquitination. Knock-down of one of the deubiquitinylases significantly impairs KSHV reactivation/replications.

The manuscript is well-written. A significant amount of data sets demonstrated the protein interactions clearly, and experiments are logically presented. Some concerns are the specificity of the OTU4's biological activity on the viral transcription. It would be important to include some cellular genes whose promoters are not regulated by K-Rta. We expect those cellular genes will not be significantly impacted by the knock-down of OTU4 or USP7. The authors showed that K63-ubiquitinylation was not changed by the overexpression of OTU4 in total cell lysates, which raises concern if the OTU4 construct is working correctly. It is helpful to include a positive control for the experiments.

Reviewer #3: In this report, the author has identified two K-Rta binding deubiquitinases (DUBs) that play a crucial role in stabilizing K-Rta and facilitating KSHV reactivation. Significantly, the author has demonstrated that OTUD4 utilizes its N-terminal domain to recruit both K-Rta and USP7, thereby enhancing the deubiquitination process and subsequently stabilizing K-Rta. This, in turn, results in the augmentation of herpesvirus reactivation. This discovery holds significant importance, as it sheds light on the deubiquitination in viral replication.

**Part II – Major Issues: Key Experiments Required for Acceptance**

Reviewer #1: Overall, the data has high quality and could support the authors' conclusion.

Reviewer #2: For Fig. 2A and Fig. 5D, please include cellular genes as comparisons to demonstrate specificity. It is important to show that knock-down of OTU4 or USP7 did not globally reduces cellular transcription activity.

Figure S3 D. It is a little strange that overexpression of OTU4 did not affect amount of cellular K-63 ubiquitinylation in cotransfection study in total cell lysates. This raises a question if the OTU4 construct is O.K. It is important to include a positive control to show that the deubiquitinylation activity for both OTU4 and USP7.

The OTU4 is primarily localizes cytoplasmic and K-Rta is a nuclear protein. The majority of experiments were performed with transient transfection except Fig. 4A. However, the Fig. 4A does not have input proteins in the same gel, which makes difficult to determine how much proportion of K-Rta or OTU4 is interacting each other endogenously. The ubiqutin pathways regulates multiple cellular signaling events including transcription, thus inhibition of transcription by knock-down of the deubiquitinylase can be due to indirect mechanisms. It is therefore important to demonstrate endogenous co-localization between OTU4 and K-Rta, and may repeat the experiment with appropriate controls.

Reviewer #3: 1. The author has successfully identified the FRD motif in the C-terminal region of K-Rta that mediates its binding to the N-terminal of OTUD4. It is recommended that the author provides a more detailed description of the strategy employed for domain mapping in Figure Legend.

2. While the use of BAC-transfected SLK cells is okay, it would be valuable if the author could also confirm the endogenous interaction in KSHV (+) cell lines, such as BCBL-1, and reactivate it using a chemical inducer, such as HDACi.

3. The identification of K-Rta K218 as the major ubiquitination site for OTUD4-USP7 is an impressive finding. The author can include it as a control.

4. In Figure 4D, it is suggested that the author consider using OTUD4 overexpression, instead of knockdown, to demonstrate that RTA-3A can abolish the interaction and the function of OTUD4.

5. Again, in Figure 5C, it would be beneficial if the author could confirm the endogenous interaction in KSHV (+) cell lines, such as BCBL-1.

6. To further elucidate the role of the OTUD4-USP7 axis in regulating K-Rta ubiquitination, the author should contemplate overexpressing USP7 in both parental and OTUD4-depleted cells, in order to show the inhibitory function of USP7 in K-Rta ubiquitination.

**Part III – Minor Issues: Editorial and Data Presentation Modifications**

Reviewer #1: 1) In Fig S5C right panel, it looks like that CYLD knockdown inhibits the infectious unit of KSHV.

2) In Fig S6C legend, the cells used here should be HEK293T-shUSP7.

3) Does OTUD4 or RTA has the USP7-interaction motif?

4) Does OTUD4 also affect other herpesviruses, such as MHV68 and EBV?

Reviewer #2: It is more convincing if the authors could provide flow cytometry data for the recombination KSHV infection.

Line 124: it should read decreased transcription (not increased transcription).

Reviewer #3: 1. For KSHV, we usually use K-Rta instead of Rta. Can the author correct it?

2. The manuscript would be improved by undergoing English editing, but it's not mandatory.

PLOS authors have the option to publish the peer review history of their article (what does this mean?). If published, this will include your full peer review and any attached files.

Reviewer #1: No

Reviewer #2: No

Reviewer #3: No
---

## [Decision Letter · Decision Letter 1]

3 Jan 2024

Dear Dr. Zhang,

We are pleased to inform you that your manuscript 'Non-canonical regulation of the reactivation of an oncogenic herpesvirus by the OTUD4-USP7 deubiquitinases' has been provisionally accepted for publication in PLOS Pathogens.

Best regards,

Fanxiu Zhu, Ph.D.

Academic Editor

PLOS Pathogens

Patrick Hearing

Section Editor

PLOS Pathogens

Kasturi Haldar

Editor-in-Chief

PLOS Pathogens

orcid.org/0000-0001-5065-158X

Michael Malim

Editor-in-Chief

PLOS Pathogens

orcid.org/0000-0002-7699-2064

Reviewer Comments (if any, and for reference):

Reviewer's Responses to Questions

**Part I - Summary**

Reviewer #1: My questions are well addressed.

Reviewer #2: The authors revised manuscript sufficiently by including controls. No further concerns from this reviewer.

Reviewer #3: The author has answered all addressed questions with new data and corrected all the typos we pointed out.

**Part II – Major Issues: Key Experiments Required for Acceptance**

Reviewer #1: (No Response)

Reviewer #2: (No Response)

Reviewer #3: The author has answered all addressed questions with new data.

**Part III – Minor Issues: Editorial and Data Presentation Modifications**

Reviewer #1: (No Response)

Reviewer #2: (No Response)

Reviewer #3: The author has corrected all the typos we pointed out.

PLOS authors have the option to publish the peer review history of their article (what does this mean?). If published, this will include your full peer review and any attached files.

Reviewer #1: No

Reviewer #2: No

Reviewer #3: No

---

## [Editor Report · Acceptance letter]

9 Jan 2024

Dear Dr. Zhang,

We are delighted to inform you that your manuscript, "Non-canonical regulation of the reactivation of an oncogenic herpesvirus by the OTUD4-USP7 deubiquitinases," has been formally accepted for publication in PLOS Pathogens.

Best regards,

Michael Malim

Editor-in-Chief

PLOS Pathogens

orcid.org/0000-0002-7699-2064